# Organizational Challenges in Healthcare Services Providers for Individuals with Autism Spectrum Disorder (ASD) Considering Personnel Turnover Rate

**DOI:** 10.3390/brainsci13040544

**Published:** 2023-03-24

**Authors:** Sayyed Ali Samadi, Cemal A. Biçak, Nigar Osman, Barez Abdalla

**Affiliations:** 1Institute of Nursing and Health Research, Ulster University, Belfast BT15 1ED, UK; 2New Breeze Autism Center (NBAC), Dihok 42001, Iraq; psikologcemal0621@gmail.com; 3Independent Researcher, Erbil 44002, Iraq; nigarrosmaan@yahoo.com (N.O.); barez.00175578@gmail.com (B.A.)

**Keywords:** turnover, organizational challenges, healthcare providers, autism spectrum disorder, low and middle-income countries (LMICs), Kurdistan Region of Iraq

## Abstract

The organizational strategy and environment of the healthcare systems influence the turnover rate among healthcare provider personnel. These critical factors have received scant attention in the literature and particularly in the healthcare systems of the Kurdistan Region of Iraq (KRI) as one low-income country with a lack of infrastructural health provision foundations. In this cross-sectional study with qualitative synthesis, the turnover rate of a population consisting of 115 (85 “74%” rehabilitation and training personnel) members of a daycare center for individuals with autism spectrum disorder (ASD) during eight years in the city of Erbil was considered. The aim was to understand the organizational strategy and its impacts on the personnel turnover rate, particularly among healthcare providers. The analysis was conducted utilizing collective data over an eight-year period of service provisions. A qualitative methodological approach was adopted to understand the different aspects of turnover among the staff. The finding indicated a general turnover rate of 87% for the entire personnel and 72% for the rehabilitation and training service providers. This rate imposes considerable costs on the system. Interview analysis also yields supporting evidence for the desire of the staff to change jobs and the lack of understanding of the turnover among system authoritarians resulting in unpredictable changes and staff displacement. Further investigations are needed to understand different contributing factors to imposed or desired turnover rates among healthcare providers in KRI. The turnover over the years has imposed different challenges for the center where the data was collected and imposes unwanted negative impacts on healthcare organizations.

## 1. Introduction

Developmental disabilities or disorders (DD) are more than a scientific concept and have a long history. These diagnoses have a wide range of characteristics and aspects that impact the individual with DD and those in contact with them. The community mental health care attention for this group includes collaborative care, aftercare, and daycare rehabilitation. The collaborative care model includes primary caregiving providers in the daycare centers to provide recognition and intervention to control common core symptoms. Autism spectrum disorder (ASD) is a DD. It is known as a brain dysfunction characterized by lifelong impacts on social communication and interpersonal interaction, as well as restricted and repetitive patterns of behavior, interests, or activities) [1]. Symptoms comprise obvious tendencies for solitary play from childhood, difficulty making eye contact, lack of interest in other people, delay and bias in verbal expression, difficulty with reciprocity and mutual verbal exchange, abnormal preoccupation with specific behaviors or activities, resistance to change, and obsession with specific objects and routines, in various level of severity and different extent [2]. These individuals, particularly those living in the community, have more significant needs at different age levels, covering various requirements such as training, education, and healthcare [3,4]. On the other hand, diverse healthcare service providers also have challenges [5]. The amount and quality of the healthcare service providers’ work and the load of duties they have to meet the repeatedly changing and updating standard of care are crucial [6].

Nevertheless, healthcare service providers’ personnel have rarely been investigated to understand the different impacts of these demands [7]. This neglect is more evident in low- and middle-income countries (LMICs). The essential element of this shortage is the absence of data on theoretical or conceptual models of understanding the role that supports personnel’s perspectives about their jobs and their associated commitments. It is reported that interacting with clients who may have lifelong disabilities such as ASD leads to psychological helplessness and mental exhaustion [8]. Finding the predictive factors contributing to reducing the turnover rate and boosting the overall satisfaction of healthcare providers in the Kurdistan Region of Iraq (KRI) seems crucial for healthcare organizations’ policymakers and managers. This study provides promising, feasible, and sustainable recommendations to address the systemic challenge of lower levels of primary motivation among healthcare service providers, which has also been reported in other studies in other LMICs [9]. It is found that factors such as recruitment, training, and loss of organizational knowledge substantially increase the overall cost for healthcare service providers in managing high employee turnover [10]. The high levels of employee turnover are associated with poorer service user outcomes and are considered to be a “contagion”, negatively impacting the workforce [11]. Job turnover and its impacts on service provisions for individuals with ASD are considered for evaluation in the current study.

There is a shortage of studies regarding job turnover in most Middle Eastern countries, particularly in the KRI. Green et al. [12] reported a similar deficit in more affluent and developed countries. They believe that regardless of the increased risk for turnover among care workers in the healthcare sector, presently, there is a shortage of studies comparing contributing factors across different disciplines (such as trainers, psychological service providers, various therapists, i.e., occupational, speech or physical therapist) and services types (i.e., daycare, rehabilitation, evaluations, and administration) to understand their experiences and contributing factors.

However, this study is the first on job turnover among the staff of a center that provides daily services in Iraq’s Kurdistan region. Additionally, the study involves searching databases such as Google Scholar, PubMed, Web of Science, and PsycINFO by utilizing additional terms reflecting job turnover in Middle Eastern countries, KRI. As a general term with the keywords, Kurdistan did not yield any studies on healthcare service providers or service providers for individuals with DD or ASD.

This study was performed to attain the following objectives: (a) to understand the impacts of the turnover rate of staff and its costs since the establishment of a daycare center for children with ASD in the KRI, (b) to understand the contributing themes related to job turnover rate among the staff in different units of the center using a deep interview, and (c) to understand the factors contributing to turnover in a daycare center for individuals with ASD.

### 1.1. Job Turnover

Personnel turnover, or simply turnover, happens when an outgoing employee replaces a new employee [13]. Higher turnover means many outgoings and arrivals of personnel. More steadiness regarding personnel replacement in a working area indicates lower turnover. Generally, turnover in a job is replacing an employee with a new one because the position is left empty, and it is measured as a percentage rate, which is referred to as the turnover rate. Turnover might be voluntary or involuntary; each type of turnover has different causes and effects [14]. Voluntary turnover occurs when an employee voluntarily decides to leave the job. Various reasons may exist for this departure, and it might happen to be a desire for a better job or a breakdown. Involuntary turnover occurs when the employer or authoritarian dismisses the employee for poor performance or inability to meet the desired goals. Turnovers are also functional and unfunctional. Turnover is functional when the employees in question are low-performing individuals in the organization. Dysfunctional turnover refers to the turnover of employees who had considerable leadership potential but left the organization. These are the worst cases of turnover and the ones that organizations need to avoid by creating a succession plan. In dysfunctional turnover, a valued employee desires to leave the organization, but the organization wants the individual to stay. Also, dysfunctional turnover is a “divergence from the equilibrium where the costs of turnover equal the costs of retention” [15].

The turnover rate in a healthcare providing system is associated with different factors, of which the organizational strategy and environment are the most important ones [16]. Different turnover rates have been reported for staff working in health services. An organization’s turnover costs include employment, training, and productivity loss costs. The minimum cost rate of turnover in the field of mental health, as Waldman et al. [17] suggested, “represented a loss of >5 percent of the total annual operating budget”. According to available data, healthcare organizations and the services industries suffer when staff members terminate their affiliation [18]. On the other hand, turnover sometimes results from organizational policies and authoritarian decisions.

Although job turnover occurs throughout different job types, available literature indicates a higher rate for mental health providers due to the other demands associated with this group of healthcare service providers [12]. In particular, the turnover rate per annum for staff working with individuals with DDs based on Mitchell and Braddock [19] and Seninger and Traci [20] has been reported to be in the range of 70.7% to 77%. Working with individuals with ASD poses particular challenges [21]. The harmful impacts of personnel turnover on clinical services quality for individuals with ASD have also been documented [22], including causing damage to the therapeutic rapport and organizational integrity.

Globally, efforts have begun to explore contributing factors and predictors of job turnover to design effective interventions to control the turnover rate. The reason is that having the desire to leave the job for different reasons inhibits the ability of optimal job performance fulfillment and consequentially negatively impacts service provision. A high turnover rate, based on Sulek et al.’s [23] findings, has a negative effect on the remaining personnel’s morale and workload, particularly when untrained staff substitute for experienced staff. Finally, frequent turnover may disrupt the continuity of services, potentially negatively impacting patients’ progress in treatment.

### 1.2. The KRI Context

The center from which the data was obtained was active in the region for eight years through advocacy and management of a parent of two children with congenital developmental disabilities. This center is one of the first daycare centers in Erbil that provides training and rehabilitation services for children with different types of DD, such as children with ASD. This center is located in the capital city of the semi-independent Kurdistan region of northern Iraq. Previously, children with ASD and other DDs in the area needed to go to neighboring countries such as Iran, Turkey, or Jordan to receive needed services. There were 13 other centers in the city at the time of the study (one governmental, two charities, and ten private). In the absence of formal national census results, on the UN estimation in 2022, Iraq’s population was @42 million, of which 17% is Kurdish (@7.1 million). Erbil’s population is reported to be 1,200,000, including displaced Yazidis and Syrian Kurdish refugees. Based on the World Bank’s estimation of a 15% rate for DDs, a population of 180,000 have a type of DD. The rate for ASD is reported to be at 3000 based on the report from the Center for Kurdistan Progress [24]. The area population is reported to be young, and the rate of individuals under 15 years old is about 35%. It is also estimated that 13% of the households have a member with a type of DD. In the absence of sufficient formal governmental education, health, and social services, the population of DD receives monthly fees payable from the government and private services from the non-governmental sector.

### 1.3. Organizational Challenges and Daycare Management Issues

As Bolton [25] indicates, service scarcity and inaccessibility are apparent challenges in Iraq’s healthcare field at a national and local level. Local governments of KRI and organizations that provide healthcare services also face different cultural and social difficulties related to negative attitudes and biases toward KRI’s developmental disabilities and service-seeking behaviors. Several organizational and management factors may have contributed to the present challenges [26]. However, the impacts of these challenges might be seen in different ways. First, the personnel’s desire to keep their position and feeling of being valued impacts the turnover rate. There are other findings regarding the lack of knowledge in the country in general about the different population’s healthcare needs and limited healthcare perspectives, along with varying problems of organization, such as feasible action plans for coping with the predicted healthcare challenges [27]. The dominant style of management in the center might be categorized as an authoritarian managing style or micromanaging, which is autocratic leadership, in which an individual has total authority, decision-making power, and absolute control over his subordinates. Employees’ ideas are rarely considered; hence, at the time of the problem, individuals are to blame rather than a flawed process, and changing the individual is viewed as a solution. This style is regarded as the core of the classical management approach [28].

The conceptual framework for this study was based on theories considering the employees’ feelings about their jobs and their feelings when they fulfilled their professional aims. Some classic conceptual frameworks consider the mental processes of the employee that can be utilized for decision-making to fulfill needs [29]. Hence, contemporary theories that interpret feelings toward a job differently are more applicable. The “Social Information Processing” theory [30] stresses the importance of the social context. This approach suggests that an opinion about the job is formed when the employee is asked about it. It also suggests that the employee’s perspective regarding the position is socially constructed. Therefore, the employee’s perspective is shaped based on external sources, such as colleagues’ comments and dominant ideas [31,32,33].

A qualitative approach was adopted to attain the present study’s aim and objective. According to Sandelowski [34], reporting frequencies and percentages are integral to the analysis process. In order to help to establish the significance of findings and numbers in qualitative studies, numbers are also used. They are also used to recognize patterns and to make analytical generalizations from the data. Qualitatively, thematic analysis was undertaken. Leininger [35] noted that themes are identified by “bringing together components or fragments of ideas or experiences, which often are meaningless when viewed alone” (p. 60). The themes were collected to form a comprehensive picture of participants’ experiences and impressions about their position in the center. General themes taken from the content of the transcribed interviews were grouped according to their content into sub-themes, and participants’ own words were used to illustrate the sub-themes. To conclude, the process of thematic analysis, based on a descriptive phenomenological approach, goes from the original data to the identification of meanings, organizing these into patterns, and writing the results of themes related to the study aim and the actual context.

## 2. Materials and Methods

### 2.1. Participants and Procedure 

Data for this study were derived from the recorded data and 115 personnel of a center for children with ASD in Erbil, the Kurdistan Region of Iraq. The data was recorded in the center’s administration electronic system, EÇOP (Turkish software for electronic program planning for children with disabilities) [36]. The data covered all the personnel working in three departments: General Services (7, 6%), Administration (9, 8%), and Rehabilitation/Training (99, 86%). At the time of the study (April 2022), 34 personnel were working in three departments of General Services (3, 9.5%), Administration (9, 28%), and Rehabilitation/Training (20, 62.5%). All of them were asked to participate in the interview via a notification on the center’s daily board explaining the study and its process. The turnover rate was also extracted from the system and based on the recorded data about the personnel collected from 2015 up to the study time (seven years). Personnel were informed about the study, and the participants provided and signed written consent. The study was conducted under the Declaration of Helsinki. In the absence of a clear national protocol, and an ethical committee in the center, the researchers adhere to the seventh revised version of WMA of the Helsinki Declaration on Medical Research involving Human Subjects issued on 19 October 2013.

### 2.2. Interview Schedule

Ten personnel (29.5%) out of 34 current staff from different sections volunteered to be interviewed. Participants were from different parts of the center: two from Administration, two from Services, three from Rehabilitation, and three from Training.

There were six questions presented in a semi-structured interview. The participating personnel had a chance to learn about the study through the interviewer’s explanation offered to them orally. The session was informal. Staff members were more talkative in the session. The interview started by reminding participants that the session would be recorded and checking that they consented. The interview schedule was followed. The interview was in the Kurdish language and of 30 to 45 min duration. The first question (How do you see your position in five years?) elicited a surprising response. A further question, “Do you like your position?” aroused additional reactions from the participants. Although they had been informed that they could stop the interview anytime, none wanted to.

## 3. Results

### 3.1. Demographic Data 

The age range of the participants was 22 to 57 (Mean = 29.6, SD = 7.9), and their education ranged from 6 years to 22 years based on the years of formal education (mainly at the bachelor’s level 22, “65%”). More demographic information regarding the sample is presented in Table 1.

### 3.2. Recorded Data on Turnover in the Center

Since its establishment, the turnover rate in the center has been a significant issue, with the highest rate in 2020 due to the COVID-19 lockdown and the fundamental changes in the center’s services provision. The turnover rate has been more settled in 2021 and is predicted to be low in 2022 due to the amount of training and expenses the center considered for updating the services in September 2021.

Based on Tesone and Pizam [37], the turnover is calculated by dividing the number of personnel at the organization that left their job or were fired during a particular year by the average number of staff employed during the same year (the overall number of personnel at the beginning of the year plus the number of personnel at the end of same year divided by 2) multiplied by 100. As an instance and to make the formulation clear, a 100% turnover rate would mean that the number of staff that left the center equals the number that the center recruits.

The current study turnover rate of (5%) is based on Waldman et al. [38] for the annual turnover rate for each year considered. Table 2 depicts this rate from the center’s establishment in 2015 to the study time. The second column presents the general personnel number for each year at the center of this table. The number of turnovers at each part of the system (three main components of available services such as cleaning, transportation, and maintenance), administration (office work such as admission and accounting), and rehabilitation parts that provide healthcare services have been presented in columns. The general turnover rate indicates turnover in the center, while the healthcare turnover rate indicates turnover for healthcare providers. The final column shows the annual healthcare turnover cost in the center’s annual budget for each year and the total rate.

### 3.3. Interview Findings

All the interviews with participants were transcribed verbatim and then translated into English for the present study. Back translation was completed on a random selection of one page (7% of 14 pages) of the transcriptions with the help of a bilingual expert in human resources. An agreement was reached on the meaning of some of the proverbs and phrases used by participants. Thematic analysis was conducted to analyze the interview contents [39]. An independent rater recategorized the personnel responses on three pages of the summarized interviews, and 95% of consensus between two independent was reported. Fourteen themes and four sub-themes were extracted from the interviews.

Table 3 shows the questions and extracted themes and sub-themes and their frequencies.

The themes and four sub-themes are presented along with personnel quotations. At the beginning of the narratives in brackets, codes have been given for each quote. The first letter indicates the gender of the parent, ‘F’ for female and ‘M’ for male. The number following shows the identification code which has been given to the interviewee. The following letter indicates the department that the interviewee belongs to. Therefore [F2R] indicates a female number 2 from the rehabilitation department.

In answering the first interview question, the extracted themes were:

Optimistic:

1st sub-theme—in a better position outside the center [M2T] “I do not see myself in this position. I think five years is too much to stay in one place. My present position has nothing to offer for five years. It has clear boundaries. I will leave voluntarily.”

2nd sub-theme—staying at the same position in the center [M6S] “This is something that I will continue until the center is open. They cannot find a better than me. I know the entire city, and I am a good driver. So, my future is related to the center. Driving and shopping are my primary duties.”

Pessimistic:

[F7T] “I cannot predict my position even for the next five days. I cannot answer this question. I have done all my best to learn more about children with Autism and their functional training approaches to leave and have my own business before they fire me.”

The quotations for the second question’s extracted theme about the feeling for the job were:

Feeling good:

[M4R] “I like my job so much. Because I think it is a humanitarian job, and I also think that this is my profession. I am trained for this position, and I have years of experience. It benefits all people, even my own children and others.”

Not very sure about the feelings:

[M1A] “I have no idea! In some aspects, it is good; in some, it is not. When I was told to get the position, I had conditions. The main point is the lack of definite rules and people who think they are beyond the regulations and law. The rules are clear, but who do I blame now when authoritarians force us to change the already approved rule? How can I have a good feeling about my job?”

Extracted themes for the third question on the condition for changing their job were for a job with reasonable payment and knowledge simultaneously, providing updated knowledge and a good salary.

Reasonable payment and updated knowledge:

[M1A] If I am offered a position in which I can learn and receive good payment simultaneously, I will leave my present job. I need to learn more. I will think about changing my job and increasing my income. I will quit if I feel that I cannot sort out things in my present post. It takes my energy and makes me feel fatigued, and I am thinking about changing my job.”

Updated knowledge:

[F5R] “If the center’s situation is not improved, I prefer to go to a position such as supervision of the rehabilitation section. I can supervise more children and learn more from my colleagues. I will leave my job for a better system with higher knowledge available.”

The extracted themes from question number four about the possibility of the improvement in the job condition were; Negative, or “Not possible,” Conditionally, or “possibly,” Positive, or “no other way except for improvement” with the following quotations.

Improvement is not possible:

[M2T] “There is an obvious wide gap between ideal circumstances and our common working area. Deep gaps that might not be crossed.”

Conditionally, “Improvement is possible.”

[F3T] “There is some ray of hope. We might be able to meet acceptable standards. At least we can try. If we are provided with more assistance.”

Positive, or “No other way except to improvement.”

[M4R] “There is no other way. We have to improve and develop. If the other people did, we also could do it.”

Two themes were extracted from question five regarding the conditions for job improvement. The quotations related to each theme were as follows:

Understanding society’s dominant culture:

[M1A] “The society in which we are living is not a democratic one. Any possible endeavor to improve our job condition must understand that the dominant totalitarian approach is applicable in our system. We need to adjust to a situation where there is no clear line between right and wrong, and the rules fluctuate to meet the needs of those who have authority to control the health-giving system.”

Being in contact with other social systems

[M4R] “We need assistance to boost our level of development in the job, provide more opportunities for being in contact with society, and receive updated knowledge. Both formal and informal aspects of the job need to be improved. We need a system to defend us against those who consider themselves above us.”

The final question about the participants’ reaction if they are dismissed from their job elicited responses that were categorized under two main themes:

Becoming upset and disappointed:

1st sub-theme quotation for those who said that they showed a reaction was:

[F5R] “I became upset a lot and never forgot about it. My feedback will be asking a question about the reason. I will talk about it to people who have the power. I may cry. Then I try to forget about it and find a new job.”

2nd sub-theme quotation for those who said that they do not react was:

[M1A] “I do not feel good about it. I do not think that it was a waste of time for me. I will hand on my position to the appointed person.”

Not being upset or disappointed:

[M6S] “I won’t feel so bad and will find a similar job in another place.”

## 4. Discussion

There were three main aims for this study. The investigation was conducted to understand the impacts of turnover of healthcare personnel since the establishment of a daycare center, after the changes of the service provider of the center in 2020, and through a sudden change of the program and reorganization of the healthcare service providers in mid-2022. Over the years, the turnover has imposed different challenges for the center where the data was collected. The calculated turnover cost is critical in health service provision, especially in LMICs with a professional and financial resources shortage. The data analysis in a round of eight years of activities of the center revealed that turnover had been an essential factor but not treated seriously. The general turnover rate of 72% for the health care providers personnel for the center imposes considerable costs such as repetitive training courses for the newly hired personnel, increased probability of showing defensive behaviors related to terminations, and reduced performance quality and optimal energy. The turnover type is not documented, but generally, it is due to unpredictable authoritarian decisions due to unclear or logically unattainable aims or objectives, changes in the general policy, and financial issues.

Turnover also negatively impacts the confidence of remaining employees [40], increases the administrative period, contributes to troublesome culture, and is considered a threat to the structure of a healthcare provider organization [41]. It was revealed that staff turnover is carried out by top-down order, and the decision of the center’s founder is based on personal impressions without reference to the predetermined organizational objectives and approved action plans [42]. It seems that healthcare system reforms are also necessary, as has happened in other LMICs. The reforms in the healthcare sector in other LMICs also stressed the existence and empowerment of non-governmental organizations, such as parental associations, as the stakeholders, not sole service consumers [43], to share their ideas about organizational decisions such as personnel turnover. A cultural shift is required towards a more consultative and devolved management style. Ultimately, this is what brought about the process of change.

Regarding the second aim of the current study, which was to understand the possible contributing themes related to job turnover increasing in different units of the center, it was evident that by improving working conditions, administrators might positively impact feelings toward the job or boost the desire for staying in the job. Managers might also be able to improve retention by reducing the turnover rate. Eventually, providing opportunities for the personnel to update their knowledge may help them develop, improve and understand the value of their position. At the center, the level of turnover reduced right after professional training. Increasing employee retention is the obvious, safest, and most effective way to lower turnover costs [38]. Waldman and Arora [17] also argue that a different methodology is needed to track personnel retention. Bearing in mind that the adopted conceptual framework also stresses the importance of coworkers’ impacts and other personnel on feelings about the job, a higher turnover rate negatively impacts an organization’s remaining personnel [33].

The third aim of this study was to understand the contributing factors to turnover in a daycare center for individuals with ASD. In this example, the study showed that controlling the incidence of extreme personnel turnover might be the rational aim of any organization to boost the level of productivity and decrease unnecessary costs. Based on the analysis of the collected data in a daycare center and interview contents, it is evident that excessive turnover is a wasteful application of an organization’s resources. The organization members must be considered the most valuable assets and backbone that require much attention [44].

In congruence with previous findings [45], current data indicated that providing training opportunities and professional training in particular areas of healthcare provision is considered a vital element for personnel desire to stay in their position and their feeling of success.

Most of the incident turnover was due to the organization’s administration issues, such as changing policies, shifting the services, cutting or redistributing resources, terminating programs, and lack of loyalty to the approved plans or a lack of employment and employer communication. As Wang and Sun [46] indicated, the poor performance of an organization increases the risk of a persisting chain of events that shape a vicious cycle; poor performance leads to a higher level of personnel turnover, further exacerbating performance problems.

In sum, healthcare centers and managers of these services need to understand turnover costs in this field and recognize job burnout and job satisfaction factors. This issue is crucial, especially in countries with limited resources, and a professional understanding of these factors is more critical. Understanding the impeding factors of job satisfaction, the predictor factors, and the process and associated risk factors at its early stages are essential to allocating sufficient internal and external support to address the problem. Positive motivational factors include providing promotion opportunities, establishing a solid internal network, and income improvement. These factors can protect the healthcare system, reduce turnover, and improve job satisfaction. Providing knowledge-boosting opportunities targeting special personnel groups, teaching adaptive coping, and enhancing resilience are also proper strategies in healthcare systems.

### Limitations

The present study’s findings need to be considered in light of its limitations. First of all, the study’s small sample size reduces its level of generalization. Hence it needs to be stressed that in-depth interview findings are not generally used for generalization-making. In-depth interviews are mainly an inductive and emergent process used to create “categories from the data and then to analyze relationships between categories” [47]. Secondly, the sample was convenient, and studies on more extensive samples and recruiting representatives for different healthcare sectors might be able to yield more relevant results. Thirdly, it should be considered that the recruited sample was from the private sector, and governmental healthcare providers might have a different perspective regarding the investigation factors in this study. Fourthly and finally, the turnover rate data was collected from the data collecting system, and the data was not collected for this study. Hence, the advantage of this investigation is that this study might be able to spot the importance of understanding the healthcare providers’ perspectives on their careers in the area and be considered as a start for further investigations.

## 5. Conclusions

The present study was conducted to understand the contributing factors of the challenges associated with ASD in LMIC. The turnover imposes unwanted negative impacts on healthcare organizations in the low resources settings. It also has to be understood that all turnover is not detrimental to the organization. The turnover, along with satisfaction, profitability, and productivity in different businesses, were reported to have a linear relationship [48]. A rate of turnover is beneficial to the organization when underperforming employees are replaced with higher-performing employees, called functional turnover [49]. This type of turnover rarely happens in healthcare systems in LMICs because of a shortage of trained workforces in the field of DDs, particularly ASD.

A high turnover rate indicates that the healthcare system forces itself to spend too much time and resources on training newly recruited personnel. Effective onboarding processes are recommended so healthcare provider systems can quickly recruit and replace any departing staff. If the turnover rate is steady and unchanged, there is a higher risk of the organization relying on poor-performing personnel in the absence of proper training and resources. Further studies are needed to understand the predictor of turnover among healthcare providers in low-resource settings with different groups of healthcare stakeholders, to understand the weight of the cultural factors in this process. Healthcare turnover rates and their relationship with various psychological, individual, and organizational factors, such as personality, work type, payment, assigned duties, administration, and leadership styles [50], are the further directions of research in this field.

## Figures and Tables

**Table 1 brainsci-13-00544-t001:** The demographic information of the sample (*n* = 115).

Variable	Frequencies and Percentage
Gender Female Male	72 (62%) 43 (38%)
Marital statues Married Single	40 (35%) 75 (65%)
Citizenship Iraqi Kurdish Non-Iraqi Kurdish	100(87%) 15 (13%)
Profession Psychology Education Rehabilitation Administration Chore Services	38 (33%) 26 (23.5%) 17 (15%) 24 (21%) 10 (8%)

**Table 2 brainsci-13-00544-t002:** The turnover rate in the center from 2015 to 2022.

	Overall Personnel Number	Services Turnover	Administration Turnover	Rehabilitation Turnover	The Overall Turnover Number for the Year	General Turnover Rate	Healthcare Turnover Rate	Health Care Annual Turnover Cost in Annual Budget
2015	23	3	0	5	8	35%	22%	25%
2016	32	0	0	9	9	28%	28%	45%
2017	31	1	0	8	9	29%	26%	40%
2018	43	0	0	11	11	26%	26%	55%
2019	54	0	0	6	6	11%	11%	30%
2020	55	2	3	19	24	44%	34.5%	95%
2021	32	3	3	9	15	47%	28%	45%
2022	38	0	2	16	18	47%	42%	80%
Total	115	9	8	83	100	87%	72%	52%

**Table 3 brainsci-13-00544-t003:** Interview questions and extracted themes and sub-themes and their frequencies.

Questions	Themes	Sub-Themes
1. How do you see your position in this center in five years?	1. Optimistic (3–43%)	In a better position (2–75%)
		In the present position (1–25%)
	2. Pessimistic (4–57%)	-
2. How do you feel about your present position?	1. Feeling good (6–86%)	-
	2. Not very sure about their feelings (1–14%)	-
3. On which condition may you change your present position?	1. A better-paid job and the possibility of receiving more knowledge (3–43%)	-
	2. A job with the possibility of receiving more knowledge (3–43%)	-
	3. A job that is related to my previous experience (1–14%)	-
4. Do you think that the work situation could be improved?	1. Negative-Not possible (3–43%)	-
	2. Possibly conditional (3–43%)	-
	3. Positive-Of course (1–14%)	-
5. How the working situation could be improved?	1. Understanding particular cultural values (4–57%)	-
	2. Extending situations for establishing contacts with other systems and more social contacts (3–43%)	-
6. How do you react if you are dismissed from your present position without prior notification?	1. Feeling upset and disappointed (6–86%)	Showing no reaction (3–50%)
		Showing reaction (3–50%)
	2. Trying to be positive (1–14%)	

## Data Availability

Data is unavailable due to privacy or ethical restrictions regarding the personnel’s ideas on the system. The center name and data remain confidential.

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
