# Peer review of "Organizational Challenges in Healthcare Services Providers for Individuals with Autism Spectrum Disorder (ASD) Considering Personnel Turnover Rate"

_brainsci, 2023, doi:10.3390/brainsci13040544_

Round 1

Reviewer 1 Report

I consider the topic of the article to be important, because it refers to a phenomenon important for the quality of care of people with developmental disorders, including autism spectrum disorders. As we know, in the group of caregivers of these people there is a phenomenon of high turnover rate. It is very negative one because people who have gained experience in this specific type of care, leave their jobs and people who do not have the relevant experience come to replace them. The authors conducted research in a group of persons who are personnel of a center for children with ASD. As the aythors stated, the aim of the study was to understand the impacts of turnover rate on staff and its costs, and understanding the factors contributing to turnover.

In order to remove existing errors and shortcomings, I propose implementation of the following changes and corrections.

1. Shortening the 'Introduction' section as it contains a lot of information not related to the topic of the article.

2. Supplementing the section 'Materials and methods' with information on the evaluation of the research project by the ethics committee (this information is missing in the text).

3. Consider whether it would not be better to present demographic data in the form of a table (page 5).

4. In the 'Discussion' section there are too few references and comparisons of the results of own research with the results of other authors.

5. The 'Conclusions' section (page 11) requires major changes. In its current form, it contains general statements that do not refer to the results obtained by the authors of the article. Therefore, I propose to shorten it, and the conclusions of the research should be presented in the form of short phrases.

6. The entire text of the article requires detailed editorial correction, as it contains some linguistic errors as well as 'technical' mistakes (e.g. in table 2, page 7 it is '3 -%43', while it should be '3-43%; page 7, line 291 is 'sub-them', it should be 'sub-theme', etc).

Author Response

 (Reviewer 1)

I consider the topic of the article to be important, because it refers to a phenomenon important for the quality of care of people with developmental disorders, including autism spectrum disorders. As we know, in the group of caregivers of these people there is a phenomenon of high turnover rate. It is very negative one because people who have gained experience in this specific type of care, leave their jobs and people who do not have the relevant experience come to replace them. The authors conducted research in a group of persons who are personnel of a center for children with ASD. As the aythors stated, the aim of the study was to understand the impacts of turnover rate on staff and its costs, and understanding the factors contributing to turnover.

In order to remove existing errors and shortcomings, I propose implementation of the following changes and corrections.

We agree with the reviewer regarding the importance of the topic for service providers in the LMICS and appreciate their time for providing useful comments and considered time.

1. Shortening the 'Introduction' section as it contains a lot of information not related to the topic of the article.

Since no nonrelated sections were highlighted we excluded lines 58 to 62 from the introduction part to consider the reviewers' comments. The excluded paragraph was as follows:

Several studies showed that the impacts of long-term service provision on healthcare service providers’ personnel could be tracked through aspects such as different forms of personnel withdrawal or dismissals, known as job turnover. Ultimately, the personnel’s impression of their job can impact (harm or improve) the quality of care provided to individuals with ASD.

2. Supplementing the section 'Materials and methods' with information on the evaluation of the research project by the ethics committee (this information is missing in the text).

We moved the paragraph under the second sub-title of “Participants and Procedure” under the Methodology section to clarify the research ethics situation in the region: (204 to 207)

3. Consider whether it would not be better to present demographic data in the form of a table (page 5).

The presented data was transferred to a table and information in the text was shortened as follows:

The age range of the participants was 22 to 57 (Mean=29.6, SD=7.9) and their education was in a range from 6 years to 22 years based on the years of formal education (mainly at the bachelor’s level 22, “65%”). More demographic information regarding the sample is presented in table 1.

4. In the 'Discussion' section there are too few references and comparisons of the results of own research with the results of other authors.

Thank you very much for the comment. We agree with the limitation of the references to justify the presented finding in the discussion. Hence, three more references were cited that are listed below:

40.   Hogh A, Hoel H, Carneiro IG. Bullying and employee turnover among healthcare workers: a three‐wave prospective study. Journal of nursing management. 2011 Sep;19(6):742-51.

41.   Aarons GA, Sawitzky AC. Organizational climate partially mediates the effect of culture on work attitudes and staff turnover in mental health services. Administration and policy in mental health and mental health services research. 2006 May;33:289-301.

42.  Amanambu RA. An investigation of the intention to leave or stay of health care professionals at St. Andrews Hospital (Doctoral dissertation, Rhodes University).

5. The 'Conclusions' section (page 11) requires major changes. In its current form, it contains general statements that do not refer to the results obtained by the authors of the article. Therefore, I propose to shorten it, and the conclusions of the research should be presented in the form of short phrases.

The conclusion section was updated and shortened. One reference  (reference 45 “Lord et al 2022” and its related paragraph was excluded).

6. The entire text of the article requires detailed editorial correction, as it contains some linguistic errors as well as 'technical' mistakes (e.g. in table 2, page 7 it is '3 -%43', while it should be '3-43%; page 7, line 291 is 'sub-them', it should be 'sub-theme', etc).

Thank you very much for the corrections. The text was reviewed from the technical aspect once again and highlighted typos were corrected.

Reviewer 2 Report

This is a very interesting paper focused on the changes in healthcare services for people suffering from Autism Spectrum Disorders.

The paper is well-written and of interest for the journal and the readers; however, several changes should be made. 

ABSTRACT

1- In the abstract section, the authors should clarify the main design of the paper. Is this a cross-sectional study with qualitative synthesis? This should be described in the abstract.

2- The last sentence is starting by "the present study might highlight". I recommend to avoid this.  This is a summary instead of a conclusion.

INTRODUCTION

1- The beginning of the introduction is mainly focused on neurodevelopmental disorders. I recommend to introduce the different types of community mental health attention to these patients, before introducing factors influencing their clinical course.

2- Section 1.2. should be described at the last part of the introduction

MATERIAL AND METHODS

1- The first part of this section should be "Participants and study design".

2- Demographic data should be described in the results section.

RESULTS

1- I recommend to divide this section into several sections and include  the main description of the sample.

DISCUSSION

1- The main limitations and strenghts should be described at the end, and numbered. 

CONCLUSIONS

1- The conclusions are too long. I recommend to summarize this section and to include future directions.

Author Response

This is a very interesting paper focused on the changes in healthcare services for people suffering from Autism Spectrum Disorders.

The paper is well-written and of interest for the journal and the readers; however, several changes should be made. 

Thank you very much for the very encouraging comment we totally agree about the importance of the topic for individuals with Autism Spectrum Disorder acre providers in the LIMICs

ABSTRACT

1- In the abstract section, the authors should clarify the main design of the paper. Is this a cross-sectional study with qualitative synthesis? This should be described in the abstract.

2- The last sentence is starting by "the present study might highlight". I recommend to avoid this.  This is a summary instead of a conclusion.

ABSTRACT

Based on the comment following sentence was added to the abstract:

In this, cross-sectional study with qualitative synthesis the turnover rate of a population consisting of 115 (85 “74%” rehabilitation and training personnel) members of a daycare center for individuals with autism spectrum disorder (ASD) during eight years in the city of Erbil was considered.

The highlighted sentence changed to the following sentences:

The turnover over the years has imposed different challenges for the center where the data was collected and imposes unwanted negative impacts on healthcare organizations.

INTRODUCTION

1- The beginning of the introduction is mainly focused on neurodevelopmental disorders. I recommend to introduce the different types of community mental health attention to these patients, before introducing factors influencing their clinical course.

2- Section 1.2. should be described at the last part of the introduction

INTRODUCTION

1.      To address the presented suggestion following sentence was added before the influence of their clinical course:

The community mental health care attention for this group includes collaborative care, aftercare, and daycare rehabilitation. The collaborative care model includes primary caregiving providers in the daycare centers to provide recognition and intervention to control common core symptoms.

2.      The sub-title was deleted and the section moved under the introduction line 81.

MATERIAL AND METHODS

1- The first part of this section should be "Participants and study design".

2- Demographic data should be described in the results section.

MATERIAL AND METHODS

1.      The information regarding the design of the study brought to the end of the 1.3 sub-section of the introduction.

2.      Demographic data moved to the first results section.

RESULTS

1- I recommend to divide this section into several sections and include  the main description of the sample.

RESULTS

This section was reorganized based on the recommendations and started with the sample demographic information.

DISCUSSION

1- The main limitations and strenghts should be described at the end, and numbered. 

DISCUSSION

I numbered the limitations (four points were mentioned) and highlighted the most important point of the study.

CONCLUSIONS

1- The conclusions are too long. I recommend to summarize this section and to include future directions.

CONCLUSIONS

The section was updated and shortened and a sentence on the further direction was added.

Round 2

Reviewer 1 Report

After the corrections and changes, the article looks much better. However, it still needs to be changed as the authors seem to have misunderstood some points. For example, in the section 'Conclusions' (page 13, lines 500-511), the authors refer to the results of other studies, which should not be present here. Comparing your own results with others should be done in the 'Disussion' section. The ‘Conclusion’ section should only contain conclusions related to your own research. Alternatively, you can mention what their originality is. Keep in mind that this section is among the most important in the article, which is why many readers start reading it first.

Besides, it seems that the tables could be presented in a more concise form. I hope this technical problem will be addressed by the editorial correction.

Author Response

Thank you for the comments regarding the organization of the conclusion. Based on the recommendation the ‘Conclusion’ section was updated and reorganized and the references were brought to the discussion section.

The references section also reorganized

Regarding the tables, no concrete recommendation was presented and a general suggestion was presented to the journal's editorial, no changes considered. For this factor we tried to follow the journal's template style.